# TOPIC-AWARE CONTEXTUALIZED TRANSFORMERS

## ABSTRACT

Training on disjoint fixed-length segments, Transformers successfully transform static word embeddings into contextualized word representations. However, they often restrict the context of a token to the segment it resides in and hence neglect the flow of contextual information across segments, failing to capture longer-term dependencies beyond the predefined segment length. This paper uses a probabilistic deep topic model to provide contextualized embeddings at both the token and segment levels. It also introduces a contextual next-word embedding guided topic attention module, injecting contextualized topic information into Transformer-based architectures. The proposed method not only captures global semantic coherence of all segments and word concurrence patterns, but also enriches the representation of each token by adapting it to its local context, which goes beyond the segment it resides in and can be flexibly defined according to the target task while maintaining control over memory footprint and computational time. Experiments on various corpora show that adding only a few extra parameters, the proposed topic-aware contextualized transformers consistently outperform their conventional counterparts, and can be used to generate coherent sentences and paragraphs.

## 1 INTRODUCTION

Language models (LMs) play an important role across a range of natural language processing tasks, such as text summarization (Rush et al., 2015; Gehrmann et al., 2018), neural machine translation (NMT) (Sutskever et al., 2014; Cho et al., 2014a), and image captioning (Herdade et al., 2019; Anderson et al., 2018; Xu et al., 2015). Existing neural LMs are often built on either recurrent units, as used in recurrent neural networks (RNNs) (Cho et al., 2014b; Hochreiter and Schmidhuber, 1997), or purely the attention mechanism based modules, as used in the Transformer and its various generalizations (Vaswani et al., 2017; Dai et al., 2019; Radford et al., 2019). Moving beyond traditional recurrent units, Transformers mainly rely on attention mechanisms, in which the direct connections between long-distance word pairs might ease optimization and enable the learning of long-range dependency (Dai et al., 2019), and have recently demonstrated state-of-the-art performances on a wide range of sequence modeling tasks.

Rather than representing a token using a predefined word embedding vector, each Transformer layer creates a contextualized representation of each token by attending to different parts of the input segment (Ethayarajh, 2019), allowing the same word to take different representations depending on its context. However, Transformers are usually trained on disjoint fixed-length segments, without any information flow across segments (Dai et al., 2019), limiting the contextualization within the current segment. Therefore, they often fail to take full advantage of many other rich contextual information, such as longer-range word dependencies beyond the segment length and semantic relationships between neighboring segments. While a naive solution to explore richer contextual information is to increase the segment length, in practice, it is usually infeasible due to limited resources, which requires $\mathcal{O}\left(N^2\right)$ for the window $N$ of inputs at each layer.

Some long-range transformer variants (Dai et al., 2019; Rae et al., 2020; Rae and Razavi, 2020) aim to extend context via compression, which use compressed memory cells for preserving the previous segments' information. The Transformer-XL (Dai et al., 2019) builds up recurrent connections between segments, concatenating the past activations with a memory cell of size M ≥ N, which results in an attention cost of $\mathcal{O}\left(N(M + N)\right)$. However the memory cell still requires a considerable space $L \times M \times d_{model}$ in a L-layer transformer with embedding size of $d_{model}$, which consumes a non-negligible cost (Rae and Razavi, 2020). Rae et al. (2020) shorten the range of attention

for Transformers by compressing the past memories into fine-grained and coarser compressed memory slots, while still suffering from memory consuming as the memory size is quiet large ($> 1000$). In addition, some efficient versions focusing on Transformer model's self-attention mechanism have also recently been explored. These models reduce memory requirements by leveraging sparsity in the attention layers (Sukhbaatar et al., 2019), exploiting a factorized sparse representation(Child et al., 2019), replacing dot-product attention with locality-sensitive hashing to decrease complexity (Kitaev et al., 2020), or using product-key attention to increase the key space (Lample et al., 2019). Besides, Chen et al. (2019) represent sentence-level context as latent topic representations by using a convolution neural network, and utilize the context representations to improve translation. However, leveraging the contextualized topic information by capturing semantic coherence via a deep probabilistic topic model, to our knowledge, has not been directly applied to Transformer before. Furthermore, compared with pre-training, fine-tuning is relatively inexpensive (Devlin et al., 2019). Nevertheless, most of the current contextualized models are trained independently on different datasets, without making good use of the publicly released pre-trained models (Radford et al., 2019; Devlin et al., 2019; Radford et al., 2018; Brown et al., 2020; Peters et al., 2018; Yang et al., 2019), paired with unsupervised pre-training on a large amount of training data. This motivates us to explore a general intervention based on those predecessors for performance gain with little computation cost, providing longer-range dependencies through a deep topic model.

Different from RNN or Transformer-based LMs, topic models (Blei et al., 2003; Teh et al., 2006; Zhou and Carin, 2015; Gan et al., 2015; Zhou et al., 2016; Zhao et al., 2018) are well suited for capturing global semantic coherency by extracting word concurrence patterns into semantically meaningful topics, which can be viewed as the contextualized word representations of the entire target corpus including all segments. Since topic models are appropriate to capture long-range dependencies, some approaches attract significant recent interest by leveraging topic models to improve RNN-based language models (Dieng et al., 2017; Ahn et al., 2016; Lau et al., 2017; Wang et al., 2018a; Guo et al., 2019). Dieng et al. (2017) and Ahn et al. (2016) integrate the syntactic dependencies of RNNs and semantic topics of latent topic models. Lau et al. (2017) introduce an attention based convolutional neural network to extract semantic topics for extending the RNN cell. Wang et al. (2018a) learn the global semantic coherence of a document via a neural topic model and use the learned latent topics to build a mixture-of-experts language model. Guo et al. (2019) extract recurrent hierarchical semantic structure via a dynamic deep topic model to guide natural language generation. Motivated by recent successes on integrating topic information into RNN-based LMs, here we focus on using topic model to provide richer contextual information for improving the Transformer. In particular, we consider using Poisson gamma belief network (PGBN) (Zhou et al., 2016; Zhang et al., 2018), a state-of-the-art probabilistic topic model which can be equivalently represented as a multi-stochastic-layer deep generalization of vanilla topic models (Blei et al., 2003; Zhou et al., 2012), to extract globally shared semantical topic representations of user-defined contexts.

To this end, three different types of contextual topic information are provided to introduce long-range semantic dependencies into Transformers. ($i$) We first introduce the contextual token embedding (TE) guided by topic model to enrich the representation of each token, which not only extracts global semantics from the corpus, but also provides localized representation of a token given either its preceding or surrounding context (which one to use is task-dependent). ($ii$) To utilize contextual information of a segment, we develop the contextual segment embedding (SE) to construct a set of virtual words, which is placed before the word sequence of the current segment and fed into Transformer. As such, the generation of any token in one segment depends on semantic context from the previous segments. ($iii$) After that, we further develop a multi-head topic attention (TA) module into the Transformer, selecting semantically related topics for generating each token, a design inspired by how a token is generated by a topic model given the topics and corresponding topic proportion. To encourage topic select-attention to focus on the topics where the predicting token is more likely to be assigned to by the topic model, during training, we add a restriction between the attention weights and the latent representation of the predicting word. Besides, a sparse penalty is employed on the topic select-attention, encouraging the network to focus on only a small subset of extracted topics. Moving beyond conventional transformers, our model can not only utilize longer-range word dependencies beyond the segment length and semantic relationships across all segments, but also generalize easily to any pre-trained Transformer-based model by jointly fine-tuning on the target corpus. It only adds minor memory and computation overhead comparing with fine-tuning the Transformer-based model alone. We demonstrate the effectiveness of our method both quantitatively and qualitatively.

## 2 PRELIMINARIES

To train Transformer-based LMs, the underlying word sequence of a corpus is usually broken into fixed-length non-overlapping segments, without any information flow across segments. We denote $\boldsymbol{s}_n = (s_{n1}, \dots, s_{nI})$ as the $n$-th segment of $I$ consecutive tokens, where $s_{ni} \in \{1, \dots, V\}$ and $V$ is the vocabulary size. The segment length $I$ is chosen to balance the ability to model long-range word dependencies with the memory and computation cost. Note the segments fed into Transformers no longer respect natural document boundaries, which means a segment could consist of the tokens from more than one document. Below we provide a brief overview of Transformers (Vaswani et al., 2017) and PGBN (Zhou et al., 2016), a multi-stochastic-layer deep topic model. To make them compatible with each other, each segment is fed into Transformer as a document analyzed in PGBN.

**Vanilla Transformer networks:** Like a standard LM, Transformers are trained by maximizing the likelihood of all segments $\mathcal{L} = \sum_n \mathcal{L}(\boldsymbol{s}_n)$, $\mathcal{L}(\boldsymbol{s}_n) = \sum_i \log P_{\boldsymbol{\Omega}}(s_{ni} \mid s_{n,<i})$, where $s_{n,<i}$ consists of the preceding tokens of $s_{ni}$ within the $n$th segment, and $\boldsymbol{\Omega}$ the parameters for modeling the conditional probability. Our proposed method can be applied to improve both Transformer encoder and decoder architectures (Vaswani et al., 2017; Dai et al., 2019; Radford et al., 2019; Devlin et al., 2019; Radford et al., 2018). For brevity, we will mainly show how to use PGBN to better contextualize through the Transformer decoder, which consists of $L$ layers as

$$Z^0 = \mathrm{WE} + \mathrm{PE}, Z^l = \mathrm{TransformerBlock}\left(Z^{l-1}\right), P(u) = \mathrm{softmax}\left(Z^L \mathbf{W}_e^T\right), \quad (1)$$

where WE and PE are the word and position embeddings of $s_{n,i-1}$ when predicting the $i$-th token of the segment, $\mathbf{W}_e \in \mathbb{R}^{V \times d_{model}}$ the embedding matrix, and $Z^{1:L}$ the outputs of all $L$ layers, each of which consists of a multi-head self-attention block followed by a feed-forward neural network (Vaswani et al., 2017). To facilitate these connections, all layers in the model, as well as the embedding layers, produce outputs of dimension $d_{model}$. See previous works (Vaswani et al., 2017; Devlin et al., 2019) for more details on Transformers.

**Deep topic model:** PGBN is used to provide semantically meaningful contextual representations to guide Transformers. We represent segment $\boldsymbol{s}_n$ as a bag-of-words (BoW) count vector $\boldsymbol{d}_n \in \mathbb{Z}_+^V$, the $v$-th element of which counts how many times term $v$ in the vocabulary of size $V$ appears at the $n$-th segment. The generative model of PGBN with $T$ hidden layers, from top to bottom, is expressed as

$$\boldsymbol{\theta}_n^T \sim \mathrm{Gamma}\left(\boldsymbol{r}, \boldsymbol{\tau}_n^{T+1}\right), \dots, \boldsymbol{\theta}_n^t \sim \mathrm{Gamma}\left(\boldsymbol{\Phi}^{t+1}\boldsymbol{\theta}_n^{t+1}, \boldsymbol{\tau}_n^{t+1}\right),$$
$$\boldsymbol{\theta}_n^1 \sim \mathrm{Gamma}\left(\boldsymbol{\Phi}^2\boldsymbol{\theta}_n^2, \boldsymbol{\tau}_n^2\right), \boldsymbol{d}_n \sim \mathrm{Poisson}(\boldsymbol{\Phi}^1\boldsymbol{\theta}_n^1), \quad (2)$$

where the shape parameters of gamma distributed hidden units $\boldsymbol{\theta}_n^t \in \mathbb{R}_+^{M_t}$ are factorized into the product of connection weight matrix $\boldsymbol{\Phi}^{t+1} \in \mathbb{R}_+^{M_t \times M_{t+1}}$ and hidden units $\boldsymbol{\theta}_n^{t+1}$ of the next layer. The global semantics of entire training corpus are compressed into $\boldsymbol{\Phi}^{1:T}$, representing topic relations of T layers. $\boldsymbol{\theta}_n^t$ denotes a local semantic representation of input $d_n$, indicating its topic proportion at $t$-th layers. See Zhou et al. (2016) for more details on PGBN.

## 3 CONTEXTUALIZED TRANSFORMERS

In a Transformer-based model, an essential step is to introduce a word embedding matrix $\mathbf{W}_e \in \mathbb{R}^{V \times d_{model}}$, the $v$-th row of which provides a $d_{model}$-dimensional representation of the $v$-th token of the vocabulary. This matrix is often pre-trained on large corpora and fine-tuned afterwards on target corpus, where each token is simply represented with its corresponding embedding vector in $\mathbf{W}_e$. Given $\mathbf{W}_e$, the Transformer architecture itself can be considered as transforming each input segment, represented as a sequence of static word embedding vectors, into a sequence of contextualized word representations (Ethayarajh, 2019), which allow the same word to take different representations depending on its context. However, the contextualization is often limited to the segment itself of a fixed length, neglecting the longer-range word dependencies beyond segment length and semantic relationships between neighboring segments. To advance the longer-context information, we consider providing richer contextual information to guide the Transformer with PGBN, which is good at extracting globally semantic topics and localized feature representation of a context. Fig. 1 (a) shows the overall architecture of the proposed model, where a basic Transformer block is in conjunction with a multi-layer topic model. Firstly, the topic model extracts the contextual representation of each token as TE, directly adding to the embedding space, and the contextual representation on segment level as SE, which is placed in front of the current segment. Then an additional multi-head topic

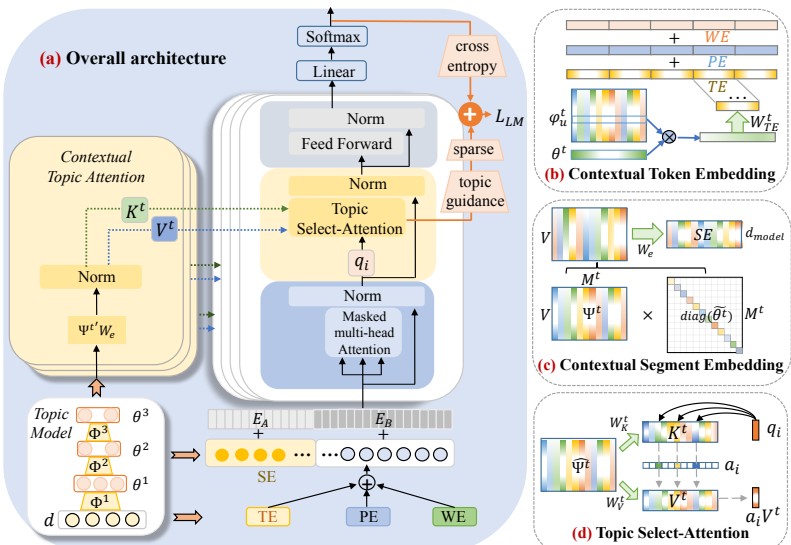

Figure 1: (a) The overall architecture of the proposed model, where the TE, SE and TA modules are highlighted in yellow color. (b) Visualization of the proposed contextual token embedding (TE) for token $u$, where WE and PE denote word embedding and position embedding. (c) Visualization of segment embedding (SE), analogized as virtual tokens and placed before the word sequence. (d) Visualization of the topic select-attention module, interleaved into transformers as shown in (a).

attention module is used to attend to different semantic topics as keys and values, according to the query calculated from the Transformers. Furthermore, the sparse regularization and a next-word prediction guidance are employed.

### 3.1 MULTI-LAYER TOPIC-AWARE CONTEXTUALIZATIONS

In this section, we will show how to integrate the hierarchical topic semantics learned with a $T$-stochastic-layer PGBN (2) into existing Transformer-based models. The topic matrix of $t$-th layer extracted from the target corpus is represented as $\boldsymbol{\Psi}^t = \prod_{t'=1}^t \boldsymbol{\Phi}^{t'} \in \mathbb{R}_+^{V \times M_t}$, containing $M_t$ topics, which tend to be more semantically specific in lower layers and become more general when moving upwards. The normalized topic proportion vector $\boldsymbol{\theta}^t = (\theta_1^t, \ldots, \theta_{M_t}^t)'$ over $\boldsymbol{\Psi}^t$ summarizes the representation of the local context of a token. Besides, we define $\widehat{\boldsymbol{\theta}}^t$ over $\boldsymbol{\Psi}^t$ as the contextual representation at the segment-level, which represents the first several segments preceding the current one, shared within the current segment.

**Contextual token embedding (TE):** To enrich the representation of each token by adapting it to both the target corpus and the local context it resides in, our first key idea is to introduce a contextual token embedding guided by a multi-stochastic-layer topic model. More specifically, we first need to define the local context of token $u$ depending on the learning task, which is composed of its preceding tokens (e.g. for text generation) or both its preceding and following ones (e.g. for text classification), going beyond the current segment $u$ resides in. We summarize the user-defined local context of $u$ into a BoW vector $\boldsymbol{d} \in \mathbb{Z}_+^V$. As shown in Fig. 1 (b), given the topic matrix $\boldsymbol{\Psi}^t$ shared globally by all segments, we use an inference network to project $\boldsymbol{d}$ to $\boldsymbol{\theta}^t$ that represents the topic proportions of $\boldsymbol{d}$ under $\boldsymbol{\Psi}^t$. We then define a localized contextual feature vector as $\boldsymbol{\psi}_{u:}^t \odot (\boldsymbol{\theta}^t)' \in \mathbb{R}_+^{M_t}$, where $\boldsymbol{\psi}_{u:}^t \in \mathbb{R}_+^{M_t}$ denotes $u$-th row of $\boldsymbol{\Psi}^t$ and $\odot$ denotes an element-wise product. In other words, the local context topic proportion $\boldsymbol{\theta}^t$ is used to re-weight topic vector $\boldsymbol{\psi}_{u:}^t$. Thus the $m$-th element of $\boldsymbol{\psi}_{u:}^t \odot (\boldsymbol{\theta}^t)' / \|\boldsymbol{\psi}_{u:}^t \odot (\boldsymbol{\theta}^t)'\|_1$ represents the probability of assigning token $u$ to topic $m$ at layer $t$. Since topics at different layers reveal hierarchical aspects of the context, we fuse the topic information from all layers together as the contextual token embedding vector, expressed as

$$\text{TE} = \sum_{t=1}^T [\boldsymbol{\psi}_{u:}^t \odot (\boldsymbol{\theta}^t)' / \|\boldsymbol{\psi}_{u:}^t \odot (\boldsymbol{\theta}^t)'\|_1] W_{TE}^t, \tag{3}$$

where $W_{TE}^t \in \mathbb{R}^{M_t \times d_{model}}$ is a projection matrix mapping the $M_t$-dimensional feature vector to a $d_{model}$-dimensional contextual token embedding vector. Note the contextualized token embedding depends on not only its position in the vocabulary, but also its local context that determines the proportions of different topics, which reflect the underlying semantics of the local context. For each token, we modify its fixed embedding vector from $\mathbf{W}_e$ by adding its topic-guided contextual embedding and the position embedding, i.e., $\text{E} = \text{WE} + \text{TE} + \text{PE}$.

**Contextual segment embedding (SE):** The second key idea of the paper is to provide a localized representation of each segment given its local context, which is defined as the first several preceding segments for text generation. More specifically, we summarize these preceding segments into a BoW vector $\tilde{\boldsymbol{d}} \in \mathbb{Z}_+^V$. Given the topic matrix $\boldsymbol{\Psi}^t$ at layer $t$, we infer the topic proportion vector of $\tilde{\boldsymbol{d}} \in \mathbb{Z}_+^V$ as $\tilde{\boldsymbol{\theta}}^t$, which serves as a contextualized embedding vector for the current segment. As shown in Fig. 1 (c), the contextual segment embedding matrix with hierarchical topic information is constructed as

$$\text{SE} = \text{Concat}[\text{SE}^1, ..., \text{SE}^t, ..., \text{SE}^T], \text{SE}^t = (\boldsymbol{\Psi}^t \, \text{diag}(\tilde{\boldsymbol{\theta}}^t))' \mathbf{W}_e \tag{4}$$

where $\text{SE}^t \in \mathbb{R}^{M_t \times d_{model}}$ denotes the segment embedding at layer $t$, each row of $\boldsymbol{\Psi}^t$ is element-wisely reweighted by $\tilde{\boldsymbol{\theta}}^t$, which is then further projected by $\mathbf{W}_e$ into the input embedding space. Each row of this contextual segment embedding matrix is considered as a localized embedding of a topic (*i.e.*, a column in $\boldsymbol{\Psi}^t$) and here analogized as a contextual virtual token. By concatenating segment embeddings across all layers, SE contains $\sum_{t=1}^{T} M_t$ virtual tokens and can be placed before the word sequence of the current segment, as shown in Fig. 1 (a). In order to distinguish the virtual token from original input token, we further add embedding $\{E_A, E_B \in \mathbb{R}^{d_{model}}\}$ respectively to help discriminate them. As these virtual words are not ordered, their embedding vectors are not combined with any position embedding. By integrating the contextual semantic information into the input space of Transformer by those virtual tokens, all real tokens in the original segment can relate to all topics with standard self-attention in the Transformer blocks. More specifically, each token in the segment is accessible to all the virtual tokens (segment embeddings) from specific to general perspective.

**Topic Attention (TA):** The third key idea of this paper is to add topic attention (TA) into Transformer layers, which is implemented with a topic select-attention block. As shown in Fig. 1 (a), for each layer of all $T$ topic layers, the topic matrix $\boldsymbol{\Psi}^t \in \mathbb{R}^{V \times M_t}$ (containing $M_t$ topics) is first projected through the word embedding matrix $\mathbf{W}_e$, reducing the dimension of each topic from $V$ to $d_{model}$. Then the projected topic vectors are then fed into a layernorm layer, following the implementation of Vaswani et al. (2017), calculated as

$$\hat{\boldsymbol{\Psi}}^t = \text{LayerNorm}((\boldsymbol{\Psi}^t)' \mathbf{W}_e) \in \mathbb{R}^{M_t \times d_{model}}. \tag{5}$$

Then we build a multi-head topic select-attention to explore the relation between the query $\boldsymbol{q}_i$ of standard Transformer and the $M_t$ topics $\hat{\boldsymbol{\Psi}}^t$, which is desired to select semantically related topics given a token. As shown in Fig. 1 (d), with transforming matrices $\mathbf{W}_K^t, \mathbf{W}_V^t \in \mathbb{R}^{d_{model} \times d_{model}}$, this $\hat{\boldsymbol{\Psi}}^t$ can be projected as keys $\mathbf{K}^t = \hat{\boldsymbol{\Psi}}^t \mathbf{W}_K^t \in \mathbb{R}^{M_t \times d_{model}}$ and values $\mathbf{V}^t = \hat{\boldsymbol{\Psi}}^t \mathbf{W}_V^t \in \mathbb{R}^{M_t \times d_{model}}$. Thus, we attend $\boldsymbol{q}_i$ into $M_t$ keys to obtain attention weights as

$$\boldsymbol{a}_i = \text{softmax}(\boldsymbol{q}_i(\mathbf{K}^t)'/\sqrt{d_{model}}) \in \mathbb{R}_+^{1 \times M_t}, \tag{6}$$

which are then used to aggregate the values into topic attention output as $\boldsymbol{a}_i \mathbf{V}^t \in \mathbb{R}_+^{1 \times d_{model}}$. This provides a natural way to leverage global semantic topics into Transformers.

(*i*) *Regularization of contextual next-word embedding:* For language generation, a common goal of learning the attention output of $\boldsymbol{a}_i$ in (6) is to better predict token $u_i$ given previous tokens $u_{<i}$. Note in a topic model, token $u_i$ chooses the $m$-th topic with probability $p_{i,m} = \frac{\psi_{u_i,m}^t \theta_m^t}{\sum_{m'} \psi_{u_i,m'}^t \theta_{m'}^t}$. Hence, in order to guide the topic select-attention with next-word embedding, we can regularize the attention weights with a loss function as $L_{i,predict} = \|\boldsymbol{a}_i - \boldsymbol{p}_i\|_2^2$, where $\boldsymbol{p}_i = (p_{i,1}, \ldots, p_{i,M})$, and the indices of heads and layers are omitted for brevity. Intuitively, we want query $\boldsymbol{q}_i$ to attend on topics where the predicting token $u_i$ is more likely to be assigned to. In addition, the attention weight vector $\boldsymbol{a}_i$ is also encouraged to be sparse with $L_1$-norm as $L_{i,sparse} = \|\boldsymbol{a}_i\|_1/\|\boldsymbol{a}_i\|_2$. The intuition behind this regularization is that a token is often only strongly associated with a small subset of topics.

(*ii*) *Integration of TM into LM:* Considering the multi-layer topics, it is reasonable to integrate them into Transformers in a hierarchical way. As Rae and Razavi (2020) remark that it is not necessary to use long-range memories at each model layer, placing them in the latter layers and interleaved across the network with equal spacing result in good performances. Sukhbaatar et al. (2019) also observe that transformers converge to using smaller attention spans for lower layers in the network, which is corresponding to the concrete topics of bottom topic layers. Thus we interleave the multi-layer topics into transformers with equal spacing from the bottom to up layers. Take a three-layer topic model as an example, the topics from layers 1, 2, 3 are integrated into layers 4, 8, 12 of the transformer (12

layers in total), respectively, through the topic select-attention module. In other words, the query at lower layers of Transformer attends to more specific topics captured by the bottom layer of PGBN, and the query at higher layers focuses on those general topics from upper layers.

Based on the proposed contextualized token and segment embeddings and a novel topic attention module, we construct topic-aware contextualized Transformers under a multi-layer topic model to capture longer-range word dependencies beyond the segment length and semantic relationships between neighboring segments. Note those three modules can extend as long context as you wish, which can be set flexibly depending on tasks, without more computation consume. Afterwards, we will describe how to jointly fine-tune the contextualized transformers with those topic interventions.

## 3.2 MODEL INFERENCE

The proposed contextualized Transformer learns topic model and language model jointly, whose loss functions are denoted as $L_{TM}$ and $L_{LM}$, respectively. For training PGBN, all segments of the target corpus are treated as BoW vectors ($d_1, ..., d_N$), ignoring word order. We introduce a Weibull hybrid autoencoding inference (WHAI) network (encoder) (Zhang et al., 2018) for PGBN (decoder). Denoting $Q = \prod_{t=1}^{T} \prod_{n=1}^{N} q(\boldsymbol{\theta}_n^t \,|\, \boldsymbol{d}_n)$, the negative ELBO of PGBN can be expressed as

$$L_{TM} = -\sum_{n=1}^{N} \mathbb{E}_Q \left[ \ln P\left(\boldsymbol{d}_n \mid \boldsymbol{\Phi}^1 \boldsymbol{\theta}_n^1\right)\right] + \sum_{n=1}^{N} \sum_{t=1}^{T} \mathbb{E}_Q \left[ \ln \frac{q\left(\boldsymbol{\theta}_n^t \mid \boldsymbol{d}_n\right)}{P\left(\boldsymbol{\theta}_n^t \mid \boldsymbol{\Phi}^{t+1} \boldsymbol{\theta}_n^{t+1}, \boldsymbol{\tau}_n^{t+1}\right)} \right], \tag{7}$$

where the weight matrices $\{\boldsymbol{\Phi}^t\}_{t=1}^{T}$ are updated with SG-MCMC as in Cong et al. (2017), and the parameters of the inference network are denoted as $\mathbf{W}_I$. By integrating the proposed topic information into an existing Transformer-based LM and adding the restrictions on attention weights $\boldsymbol{a}_i$, the loss $L_{LM}$ over a set of training examples $\mathcal{U} = \{u_1, ..., u_I\}$ is defined as

$$L_{LM} = -\sum_i [\log P\left(u_i | u_{<i}; \boldsymbol{\Omega}, \{\boldsymbol{\Phi}^t, \boldsymbol{\theta}^t, \tilde{\boldsymbol{\theta}}^t\}_{t=1}^{T}\right) - L_{i,sparse} - L_{i,predict}], \tag{8}$$

where $\boldsymbol{\Omega}$ represents the Transformer-related parameters. Therefore, the final training objective is a linear combination of $L_{TM}$ from the PGBN based topic model and $L_{LM}$ from the Transformer-based LM, which is minimized to estimate $\{\boldsymbol{\Omega}, \mathbf{W}_I, \{\boldsymbol{\Phi}^t\}_{t=1}^{T}\}$. More details are included in Appendix A.

## 4 EXPERIMENTAL RESULTS

We first provide quantitative comparisons on two different natural language processing tasks, and then qualitative analysis to illustrate how the proposed contextualizations help improve Transformers. To verify the effectiveness of our method, we integrate topic-based token embedding (TE), segment embedding (SE), and topic attention (TA) into existing pre-trained Transformer models, fine-tuning from the released checkpoints. For each task, our method shares the same model architecture as the baseline. All models are optimized and evaluated on a single 2080Ti GPU within a few hours. We use the Adam optimizer (Kingma and Ba, 2015), where the experimental settings remain the same as baseline models provided by the authors. We use $[M_1, M_2, M_3] = [100, 80, 50]$ as the number of topics in a three-layer PGBN, and set the hyper-parameters as $\boldsymbol{r} = \mathbf{1}, \boldsymbol{\tau}_n^t = 1$. Python code is provided in the Supplement.

### 4.1 QUANTITATIVE COMPARISON

**Language generation**   We choose GPT-2 (Radford et al., 2019) and Transformer-XL (Dai et al., 2019) as baseline LMs. GPT-2 is realized by pre-training a Transformer decoder and then fine-tuning on each specific task. Transformer-XL introduces a segment-level recurrence mechanism to learn dependency beyond a fixed length without disrupting temporal coherence. We use perplexity as the evaluation metric and consider three publicly available corpora, including WikiText-103 (WT103) (Merity et al., 2017), WikiText2 (WT2) (Merity et al., 2017), and Penn Treebank (PTB) (Mikolov and Zweig, 2012). WT103 and WT2 contain 103M and 2M training tokens from Wikipedia articles, respectively, and word-level PTB has only 1M training tokens. Given the pre-trained GPT-2, we fine-tune contextualized GPT-2 on each of these three datasets, with the same vocabulary, tokenizer and experimenting settings as used in GPT-2. Different from GPT-2, Transformer-XL is trained on each dataset respectively, where the authors only provide a pre-trained model on WT103 but not on PTB and WT2. Since WT103 is the largest available word-level language modeling benchmark with long-term dependency (Dai et al., 2019), it is feasible to use the pre-trained model on WT103 as our baseline to fine-tune on three datasets. Both the preceding segment window sizes of TE and SE are set as 4 for text generation. We conduct ablation studies to examine the effects of three proposed

Table 1: Perplexity of different models (lower is better ).

| Model | GPT-2-base | | | | Transformer-XL-Large | | | |
|---|---|---|---|---|---|---|---|---|
| | # Param | WT103 | WT2 | PTB | # Param | WT103 | WT2 | PTB |
| baseline + fine-tune | 117M | 16.33 | 14.66 | 15.22 | 257M | 18.30 | 17.86 | 33.71 |
| + Token embedding (TE) | 117+5.24M | 16.15 | 13.98 | 15.08 | 257+1.07M | 17.87 | 16.26 | 32.67 |
| +Segment embedding (SE) | 117+5.07M | 16.10 | 13.92 | 14.98 | 257+0.83M | 17.90 | 16.83 | 32.65 |
| +Topic attention (TA) | 117+5.81M | 16.01 | 13.92 | 15.00 | 257+2.07M | 17.86 | 16.30 | 32.64 |
| + TE + SE + TA | 117+5.98M | **15.82** | **13.67** | **14.92** | 257+2.31M | **17.84** | **16.23** | **32.60** |

Figure 2: (a) (b) Comparisons of test perplexity as a function of fine-tuning time on WT2 based on GPT-2 and Transformer-XL (T-XL). (c) Visualizing of attention weights with different regularization.

modules: TE, SE and TA. As shown in Table 1, all three contextualization techniques improve both GPT-2 (BPE token-level perplexity) and Transformer-XL (word-level perplexity), combining the three techniques together leads to the best performance for both GPT-2 and Transformer-XL, only adding slightly more parameters. More evaluation on model varieties are shown in Appendix B.

We further display in Fig. 2 (a)(b) how GPT-2, Transformer-XL, and their contextualized versions behave during fine-tuning, by showing the perplexity on the WT2 test set over time. Obviously, while GPT-2 and Transformer-XL behave well during the early stage of training, both of them show a clear trend of overfitting as the training progresses. This overfitting trend is especially concerning in Transformer-XL, although it is designed to utilize the contextual information across segments. Using the proposed contextualization methods, it takes much less time to fit data well and exhibit strong resistance against overfitting.

**GLUE** The General Language Understanding Evaluation (GLUE) benchmark (Wang et al., 2018b) is a collection of diverse natural language understanding tasks. To validate the efficiency of integrating semantic topics, we finetune the pre-trained 12-layer-Bert model on each dataset. For Glue tasks, we integrate topic semantics extracted from each input sentence. Thus SE and the next-word guided regularization $L_{predict}$ can be neglected. We use batch sizes $\in \{16, 32\}$, fine-tune for 10 epochs and perform early stopping based on each task's evaluation metric on the dev set. The rest parameters remain the same as pre-training.

Table 2: GLUE Development and Test results, scored by the evaluation server.

| Data | System | MNLI-(m/mm) acc 392k | QQP F1 363k | QNLI acc 108k | SST-2 acc 67k | CoLA mc 8.5k | STS-B pc 5.7k | MRPC F1 3.5k | RTE acc 2.5k | Average |
|---|---|---|---|---|---|---|---|---|---|---|
| Dev | **Bert-base** | 84.7/83.4 | 70.2 | 88.4 | 93.0 | 52.3 | 78.7 | 86.5 | 66.4 | 78.2 |
| | +TE+TA | 84.3/83.6 | 70.6 | 88.7 | 93.6 | 52.3 | 79.0 | 87.5 | 68.2 | 78.6 |
| Test | **Bert-base** | 84.5/83.4 | 69.6 | 90.4 | 93.4 | 52.1 | 83.1 | 88.9 | 66.4 | 79.1 |
| | +TE+TA | **84.6/83.8** | **70.0** | **90.8** | **93.7** | **53.1** | **83.8** | **89.1** | **69.6** | **79.8** |

Both the dev and test results are shown in Table 2. It is clear that adding token embedding (TE) and topic attention (TA) into Bert outperforms baseline on different tasks, especially on small datasets. Take the RTE (2.5k) for example, there is 3.2% accuracy improvement over baseline, alleviating instability on small datasets. In a word, the tasks of GLUE benefit from the contextualized Transformer architecture, extracting globally shared semantic topics of input sentences and localized feature representations for each token.

## 4.2 QUALITATIVE ANALYSIS

**Efficiency of the regularization** To verify the efficiency of our proposed regularization, we visualize the topic attention weights of a randomly sampled word with different regularization terms. Shown in Fig. 2 (c), compared with the unconstrained example (the bottom row), sparse regularization leads sparsity into the attention vector. We also find the learned attention vector would focus on some formerly unnoticed topics by applying next-word topic guidance regularization. In the top row, it is clear that the learned attention vector attend to the topics related with the predicted word and its context, while preserving sparsity. This underscores the effectiveness of our introduced regularization.

> Topic-Layer1 : investment director university course situation live values contributions futures
> Topic-Layer2 : securities director investors trust office marketing owns receive worth live
> Topic-Layer3 : million securities director investors franc trust year earnings marketing cents
>
> **Generated sequences guided by topics :**
> A good place to invest is anywhere when you have more money. It usually comes to being able to use all the tools available to you, whether it's your skills, your experience, your experience or your skills for long term investment. When you have enough money, investing can help you to do a lot. You must be able to spend the money the best you can, to do that, even if your investment will have to be a few times more important than your ability to reinvest. Most investors believe that they are at their greatest power once investments become much smaller, however that opinion is wrong because at best you're able to invest when you're just starting out. Sometimes that power is very strong, and at worst it just doesn't exist.
>
> **Preceding context** : An anarchist was arrested last week on charges of attempting to commit murder of three people in a crowded bar in Cambridge, England. The murder has sparked widespread outrage and protests which in recent days have prompted an investigation into the incident.
>
> **Generated sequences given preceding context :**
> It is thought that the gunman who stabbed three people and injured two others was radicalised by the group after attending an anti-capitalist rally in Cambridge a week previously. Police and campaigners say there is mounting evidence that he had links to extremist Islam and had travelled to the Middle East before joining the extremist movement. Two of the victims had been friends of Mr Corbyn and had been invited to join demonstrations to demand the end of British. Mr Corbyn said he was not present at the time of the event, but is believed to have been at the scene. Police are investigating the motive and motive of the man, said to be in his early 20s, who allegedly tried to commit murder.

Figure 3: The generated sequences guided by multi-layer topics and preceding context. Words in the same color are semantic-consistent. The generated texts successfully capture both syntactic and semantics.

**Text generation given topics or preceding context** Given the learned contextualized Transformer based on GPT-2, we can sample sentences conditioning on the topics from different layers. Shown in Fig. 3, the generated sequences are guided by a combination of topics at different layers, which is highly related to the given topics in terms of their semantics. These observations indicate that the contextualized Transformer has successfully captured syntax and semantics simultaneously for language generation. Besides, we visualize the generated sentences conditioning on the preceding context, by integrating the encoded hierarchical topic representations of preceding context. Interestingly, we find the generated sentences successfully capture semantics and generate semantically-related words, which may not exist in the original document (highlighted with the same color). This phenomenon is also observed in Guo et al. (2019), which might be attributed to the introduction of semantic topics. More generated samples with longer preceding context are provided in the Appendix F, where we find our proposed model can memorize longer-range context than baselines.

**Topic attention between words and topics** To further illustrate the relationship between a word and its selected topics within the topic attention module, Fig. 4 takes the word "market" as an example and visualize its attended topics of 12 heads at different layers. At each layer, we find the word is aligned to different topics, where each attention head potentially focuses on different aspects of the input word. Specifically, the attended topics of "market" are semantically related to the 34-th topic ("billion, $, bank") and the 26-th topic ("data, technology, stock") at layer 1 and so as in the upper layers, suggesting the efficiency of our proposed topic select-attention. In addition, we also find there are several heads attended to the same topics, indicating those topics might be more helpful for predicting the target word. In other words, the regularization term of our topic select-attention module encourages the model to attend on its corresponding topic while keeping its variety.

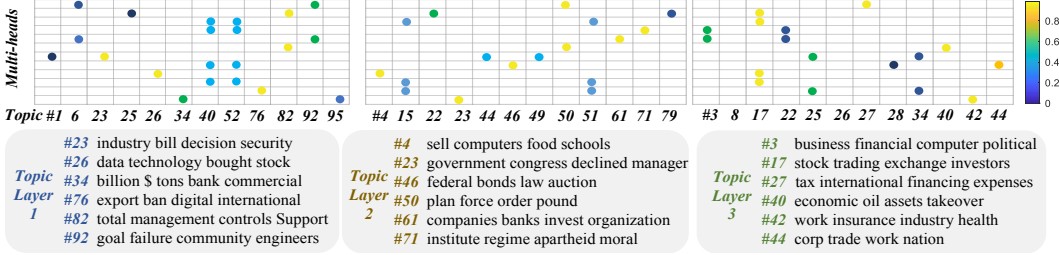

Figure 4: Visualizing the attended topics of word "market" at layers 1, 2 and 3, respectively. X-axis denotes the index of attended topics, and y-axis the index of heads. Each row denotes the most activate topic of corresponding head and we omit the other inactivate topics. Several top words of corresponding topics are listed at the bottom.

## 5 CONCLUSION

We introduce contextualized embeddings at both the token and segment levels to enrich longer-term dependencies beyond the fixed segment and semantic relationships across all segments of Transformer-based language models. Furthermore, to inject contextualized topic information into attention mechanism of Transformer-based architectures, a novel topic attention module, is further introduced. Experiments conducted on publicly available corpora demonstrate that the proposed topic-aware transformers outperform their conventional counterparts, providing better contextualized word representations for downstream tasks, and can generate coherent sentences and paragraphs conditioned on the designated multi-layer topics or preceding context.

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
