# OpenReview forum: "Topic-aware Contextualized Transformers"
_ICLR.cc/2021/Conference — Reject_

### Official Review · AnonReviewer2 · 2020-10-25
**Official Blind Review #2**

**Rating:** 4
**Confidence:** 4

**Review:**

Summary:

This paper introduces a global topic model into the Transformer to enrich longer-term dependencies beyond the fixed training segment, including contextual token/segment embedding (TE/SE) and Topic Attention (TA). However, some components seem to be unnecessary: The function of “+TA+SE” is very trivial by comparing “+Topic attention (TA)” with “+TE + SE + TA” in Table 1. In addition, the experiments of GLUE in Table 2 are only conducted on “Bert-base” with marginal improvements.

---------------------------------------
Strength:

1. This work is the first to integrate the contextualized topic information via a deep probabilistic topic model into the Transformer architecture.

2. Three different types of contextual topic information are provided to capture long-range semantic dependencies into the Transformer models.

3. Experiments on various corpora show the effectiveness of the proposed methods, with only a few extra parameters.

---------------------------------------
Weakness:

1. The ablation study is weak. TA and SE seem redundant in Table 1 compared with TA. The results in Table2 are marginal, given that the BERT-base baseline is relatively sensitive to hyperparameters. More language models (maybe larger) should be tested. Moreover, in view of (a), TE may be redundant together with TA. Since the results in Table are based on TE+TA, there should be a TE+TA comparison in Table 1.

2. “Topic attention between words and topics” in the analysis seems farfetched because the attended topics of “market” are more than 34-th and 24-th. There are more similar but less relevant topics like 82-th, 50-th, etc. According to the examples, the topic information could be related to the semantic similarity of words. There might be a risk that the topic information would have already been entailed in the pre-trained language models.

3. No experiments show the effect of hyperparameters choices from the topic model, such as the number of layers of PGBN as well as the topic number of each layer.

4. Since this work is motivated to capture longer-range dependencies, it is not clear how the topic information helps. A comparison of the model performance for different lengths of input sequences would be helpful.

5. There is no comparison with other methods about long-range transformer variants, which focus on the transformer model’s self-attention mechanism mentioned in Line 3, Page2.

---

### Official Review · AnonReviewer1 · 2020-10-26
**Not significant improvement over baseline**

**Rating:** 4
**Confidence:** 3

**Review:**

The paper introduces a novel LM architecture that combines a Transformer with a PGBN topic model, enabling the transformer model to make use of addtional context topic information. The PGBN extracts topic information from the input, which is then used to enrich the information that is available to the transformer.
They propose three different methods of incorporating this context information from the topic model into the transformer: Topic embedding vectors to add to each token, segment embeddings to summarize preceding segments, and topic attention.
The topic model intervention can be applied to available pretrained transformers without the need to retrain the models from scratch. Experiments using pretrained GPT-2 and BERT show that incorporating topic information outperforms the respective baseline transformers both in language modeling perplexity and on the GLUE benchmark.

To my knowledge this is a novel model design. The information extracted by the topic model is provided to the transformer in three different ways, ensuring the utilization of all levels of topic information. It allows for topic information to be drawn from both the previous and the current segments of the input data.
Moreover, the topic model augmentation can be applied to pretrained transformers. This makes it very versatile and greatly improves its usefulness in practice.

However,

The presented work is motivated by the fact that transformers are limited in the length of the input they can condition on, and that topic information from previous segments could alleviate this problem. The model is then evaluated on GLUE, which is not a test of long-term dependencies.
It remains unclear whether providing topic information of preceding segments is enough to allow the model to draw information from these segments that is useful for a task, beyond mimicking their style.

The authors propose to use the learnt topics to perform conditioned topic-specific text generation. Given the multi-layer topic model architecture and the fact that GPT-2 uses BPE tokenization, splitting words into subwords, this raises the question whether most topics are interpretable enough to use them in a meaningful way for conditioning manually..

Moreover, I do not find the results convincing that this methods works. I do not think that proper emphasis has been put on the baseline as it is diverging from its initial solution.

In general, I find this paper hard to read and keep track of what is being proposed. I find that it convolutes simple concepts and it could be a lot easier to understand the proposed method if written in a different way.

I think the 54 citations in the introduction is unnecessary, please keep your paper more concise on what exactly you are building on top of and what you're proposing.

I don't fully understand the TE embeddings. Is it N-gram distributions that you make a topic model over?

Is WE a "normal" word embedding?

Please elaborate on E_A and E_B

---

### Official Review · AnonReviewer3 · 2020-10-29
**This paper introduces an interesting idea of enhancing the contextualised word embedding learned by Transformers with long range semantic dependencies via topic learned by Poisson Gamma Belief Network (PGBN), a deep topic model.**

**Rating:** 7
**Confidence:** 4

**Review:**

This paper introduces an interesting idea of enhancing the contextualised word embedding learned by Transformers with long-range semantic dependencies via topic learned by Poisson Gamma Belief Network (PGBN), a deep topic model. To leverage the topic information to guide the learning process of transformers, the authors proposed two types of topic-ware embeddings and one topic attention mechanism. The experimental results show incorporating topic information can further improve the performance of Transformers.

Overall, it is an interesting paper. I would lean towards accepting it. I like the idea of injecting topic information into neural language models, like transformer. The ablation study shows that the performance of base models increases with the topic information.

Pros
1.  Capturing longer-range term dependency in Transformers is an important problem. Although there are existing works on increasing the size of the input, such as Hierarchical transformers, the authors propose to used deep probabilistic topic model to leverage the semantic information via latent topics.
2. The paper provides both quantitative results on both language generation and GLUE tasks and qualitative analysis, both show the necessity of considering topic information in transformers. The ablation study on the three components used to inject topic information is good
3. The paper is written pretty well, and easy to read.

Cons:
1. Although the ablation study is good already, I would suggest the authors conduct the following ablation studies wonder 1) how the regularisation terms would affect the performance? 2)  will the interleaving discussed on page 5 have an impact on performance?
2. Figure 4 visualises the topic attention mechanism. What are those frequently attended topics?

---

### Decision · Program_Chairs · 2021-01-07
**Final Decision**

**Decision:**

Reject

**Comment:**

This paper proposes enhancing contextualized word embeddings learned by Transformers by modeling long-range dependencies via a deep topic model, using a Poisson Gamma Belief Network (PGBN). The experimental results show incorporating topic information can further improve the performance of Transformers. While this is an interesting idea, reviewers pointed out some weaknesses:
- GLUE evaluation is not a test of long-term dependencies, it remains unclear whether providing topic information of preceding segments is enough to allow the model to draw information from these segments that is useful for a task.
- The improvement over the baseline does not seem to be significant.
- The ablation study could be improved and more experiments could be done to understand the effect of hyperparameters choices from the topic model, such as the number of layers of PGBN as well as the topic number of each layer.
- A comparison of the model performance for different lengths of input sequences would be helpful.
- There are many recent methods for long.range transformer transformer variants, it would be interesting to compare them against the proposed latent topic-based method.

Unfortunately, no answers are provided by the authors to the questions asked by the reviewers, which makes me recommend rejection.